# Systemic Lidocaine Infusions for Pediatric Patients with Cancer-Related Pain

**DOI:** 10.3390/children9121934

**Published:** 2022-12-09

**Authors:** Emily Rav, Rachna Sheth, Ali H. Ahmad

**Affiliations:** 1Pediatric Oncology Fellowship Program, Department of Pediatrics, The University of Texas MD Anderson Cancer Center, Houston, TX 77030, USA; 2Department of Pediatrics-Patient Care, The University of Texas MD Anderson Cancer Center, Houston, TX 77030, USA; 3Section of Pediatric Critical Care, Department of Pediatrics, The University of Texas MD Anderson Cancer Center, Houston, TX 77030, USA

**Keywords:** pediatrics, children, cancer, oncology, pain, lidocaine, analgesia

## Abstract

Pediatric patients with cancer experience significant distress from both treatment and cancer-related pain. Careful selection of an analgesic regimen should be based upon individual patient factors, including the level of pain, response to therapy, and physiologic profile. Refractory pain is a therapeutic dilemma frequently encountered in the pediatric cancer setting. Systemic lidocaine infusions have been described as both safe and efficacious, as prior studies show patients reporting decreased pain scores and improved quality of life after lidocaine treatment. Given the favorable side effect profile of lidocaine, it has the potential to be considered for analgesia in the setting of refractory pain. This review discusses the use of systemic lidocaine infusions for analgesia in pediatric oncology patients with cancer-related pain.

## 1. Introduction

Among pediatric patients with cancer, pain carries significant morbidity before, during, and after treatment into survivorship [1]. In addition to the cognitive stress which accompanies the journey of a cancer diagnosis, unrelenting pain presents yet another source of distress to patients and their families. Cancer-related pain can be nociceptive, originating from the tumor itself, or neuropathic, which can cause a cluster of symptoms from a lesion in the somatosensory nervous system [2,3,4]. Apart from cancer itself, procedural interventions, such as thoracentesis, placement of chest tubes, central venous catheters, urinary catheters, and nasogastric tubes, can also cause pain [5]. Pain not only limits activity and causes discomfort during therapy but can also affect patients’ perceptions of quality of life even after their cancer treatment has been completed [3]. Specifically, pain has been found to be related to mood and psychological distress in this patient population [6] and is among the top concerns of pediatric patients with cancer [2].

Treatment for cancer-related pain in pediatrics is based upon three important considerations and goals published by the World Health Organization. These include preventing pain, maximizing ease of administration of pain control, and the ability of treatment options to be tailored to the patient [4,7,8,9]. Opioids are commonly employed to treat pain in the pediatric oncology setting, as they target receptors both inside and outside the central nervous system (CNS) [10,11]. While opioids can be effective at treating nociceptive pain, neurotoxic chemotherapy can result in neuropathic pain refractory to opioids [10,11,12]. Furthermore, there are several known complications, risks, and adverse side effects of opioid use in children, including ileus, histamine release, nausea, and respiratory depression [2]. To avoid these risks, opioids are often started at lower doses in the pediatric setting, which can lead to delayed treatment and under-treatment of pain. Even when opioids are appropriately titrated, increasing tolerance can still result in refractory pain with a limited efficacy of analgesia while also accruing an increasing risk of side effects. This is especially common in patients with advanced-stage cancers.

Gabapentinoids are frequently used to treat neuropathic cancer pain; however, these drugs are only available by mouth, which precludes their use in cases of severe mucositis following chemotherapy [12,13]. Similarly, while nonsteroidal anti-inflammatory drugs (NSAIDs) are a class of commonly prescribed analgesics in other populations, previous literature has described the role of cyclooxygenase-2 (COX-2) in platelet function and graft healing. NSAIDs, which reversibly inhibit COX-2, are often avoided in the pediatric oncology population due to chemotherapy-induced thrombocytopenia and in the perioperative period after the resection of certain solid tumors due to impairment of graft healing [14,15,16,17].

In the pediatric oncology population, ketamine is a useful analgesic adjunct that acts as an NMDA-receptor antagonist. While its side effect profile does not include respiratory depression or physical dependency that is associated with opioids, ketamine is associated with other adverse effects, including hallucinations, anxiety, stimulation of the cardiovascular system, and vomiting [5,18]. Additional adjuvants, such as serotonin/norepinephrine reuptake inhibitors and tricyclic antidepressants for neuropathic pain, cyclobenzaprine and benzodiazepines for muscle relaxation, and topical analgesics can also complement an existing analgesic regimen [5,18]. Epidural and peripheral nerve blocks have also been employed routinely in certain surgical procedures and when other interventions have failed to adequately relieve pain [5,18]. Although rarely used, intravenous lidocaine infusions have been shown to be efficacious in pediatric patients with cancer and will be the subject of discussion in this review.

## 2. Systemic Lidocaine Infusions

Lidocaine is an amide local anesthetic that is approved for use as an intravenous injection [19]. It works primarily by blocking voltage-gated ion channels in neurons in the spinal cord. This affects the discharge frequency and action potential conduction speed of nerve fibers and accounts for its analgesic and anti-hyperalgesic effects [19,20,21,22,23]. It is also thought to have anti-inflammatory properties in reducing circulating inflammatory cytokines, such as interleukin (IL)-6, IL-1β, and tumor necrosis factors [22,23]. At higher doses, lidocaine also acts on NMDA receptors and suppresses spontaneous impulses generated from injured nerve fibers, thereby downregulating pain transmission pathways, which may contribute to its ability to provide analgesia beyond the length of the infusion [21,24].

The practical application of intravenous lidocaine has been described in previous literature [24,25]. In the perioperative scenarios, anesthesiologists have supervised the initiation of intravenous lidocaine in a monitored environment, such as a post-anesthesia care unit (PACU) [25]. In the pediatric hematologic and oncologic settings, intravenous lidocaine has been initiated on the inpatient floor, with certain patients receiving subsequent infusions in the outpatient clinic setting [24]. For the initiation of systemic lidocaine infusions, it is recommended to perform neurologic assessments of the level of sedation and to have cardiopulmonary monitoring readily available per patient-controlled anesthesia standards [21,24,25]. The clinical setting of intravenous lidocaine infusion will be dependent on an individual patient’s physiologic profile, variable institutional policies, and the indication for which systemic lidocaine is being infused. Table 1 summarizes medical and surgical indications for systemic lidocaine infusions that have been described in both adult and pediatric literature [22,23,24,25,26,27].

The careful selection of an analgesic regimen should be based on individual patient factors, including the level of pain, response to therapy, and physiologic profile. Therefore, it is equally important to recognize the contraindications for the administration of systemic lidocaine infusion. Table 2 summarizes contraindications for systemic lidocaine infusions that have been described in previous literature [24,25,26].

The safety and efficacy of systemic lidocaine infusions in pediatrics have mainly been evaluated using retrospective reviews and case studies. However, one prospective, randomized, double-blinded study in children was performed by Batko et al. [21]. Forty-one children undergoing major spine surgery were randomized to either a lidocaine or placebo group. The demographics were not significantly different between groups, and the two groups otherwise followed an identical perioperative protocol. In the lidocaine group, intravenous lidocaine was given as a bolus of 1.5 mg/kg over 30 min, followed by a continuous infusion at 1 mg/kg/h for 6 h post-operatively. The patients who received lidocaine had a significant reduction in morphine requirements during the first 24 h after surgery, as well as throughout the hospital course. The children who received lidocaine also advanced diet and ambulated earlier. All these findings were statistically significant.

A retrospective study by Mooney et al., evaluating 15 adolescent and young adult patients with chronic and refractory pain, established the wide range of analgesic indications for intravenous lidocaine infusions [28]. The indications for infusions included headache, neuropathy, joint pain, and sickle cell disease-related pain. The number of lidocaine infusion sessions for each subject varied from 1 to 13, and the infusions commonly included adjunct intravenous magnesium. All patients had tried numerous treatments prior to starting lidocaine. Investigators measured pain scores (scale of 1–10) and used a Likert scale for adverse effects. They concluded that the amount of pain relief was not affected by the type of pain being treated, and the side effects were minimal, with the most frequent effects reported being dizziness and numbness/tingling. Importantly, there were no correlations between the rate of lidocaine infusions and the occurrence of adverse events.

In another single-center retrospective observational study by Ayulo et al., the safety and efficacy of intravenous lidocaine were evaluated in pediatric patients with status migraines [27]. Twenty-six patients were treated, and pain scores and side effects were recorded and analyzed. In general, the infusions were well tolerated, with one interruption reported due to chest pain and anxiety. In that case, when the infusion was restarted after symptom resolution at a slower rate, there was no recurrence of those side effects.

Additionally, neuropathic pain is a common effect of both the cancer diagnosis itself and the side effects of chemotherapy and surgery. Achieving neuropathic pain relief can be difficult. Subcutaneous and intravenous lidocaine administration has been shown to help alleviate cancer-related neuropathy in some cases [29,30]. In a retrospective analysis of the tolerability and efficacy of intravenous lidocaine to treat neuropathic pain, efficacy was seen in 65% of patients [31]. Sixty-nine patients, aged 12–91 years, were all given an intravenous dose of lidocaine of 500 mg over 30 min; 5 of these patients were classified as having malignant cancer pain. Patient response was reported on a pain scale from 1 to 10. Based on the patient response, the treating team decided whether to continue the dose, adjust the dose, or discontinue the use of lidocaine. Most patients required a dose reduction due to side effects. However, on subsequent infusions, when these patients were given a reduced dose, they still experienced pain relief, so the authors concluded that lidocaine was still efficacious at the lower doses.

Dosing for systemic lidocaine infusions has varied among reporting institutions in the literature, and there is no established standard dose in pediatrics. As noted above, lidocaine does seem to be well-tolerated, with minimal adverse events and minimal correlation between dose and occurrence of side effects. In some institutions, lidocaine was first administered as a bolus (2.9 ± 0.18 mg/kg), followed by an infusion at a mean rate of 1.29 ± 0.2 mg/kg/h [27]. In the Mooney et al. study, including adolescent patients with different indications, lidocaine was given in infusions running over 2 h for headache and 6 h for neuropathic pain [28]. Dosing ranged from 40–60 μg/kg/min regardless of infusion duration. In a retrospective review of lidocaine as analgesia in cancer-related refractory pain in children, [24] lidocaine infusion dosing consisted of a loading dose of 1–2 mg/kg over 30 min followed by a continuous infusion of 1–2 mg/kg/h, not to exceed 300 mg/h.

A common dilemma when administering a medication to pediatric patients without a well-established dose recommendation is serious adverse effects. Fortunately, few serious side effects have been reported, and most patients who experienced them had other confounding characteristics. One patient experienced orthostatic hypotension requiring a fluid bolus and brief epinephrine infusion in the intensive care unit [24]. In this case, the patient had hypothyroidism and an elevated body mass index, so the authors hypothesized that she may have received a higher than intended dose of lidocaine as the dosing was based on her actual body weight as opposed to her ideal body weight.

Based on animal studies and adult patient trials, there is a known risk for seizures in pediatric patients receiving lidocaine infusions; however, the dose that lowers the seizure threshold is unknown. Berde et al. extrapolated from previous data that an infusion rate > 2 mg/kg/h or plasma concentrations of lidocaine > 6 μg/mL could result in an elevated risk of seizures [32]. In the Ayulo et al. study using lidocaine infusions to treat status migraines, one patient with developmental delay and a seizure disorder suffered a seizure and subsequent ictal asystole while receiving intravenous lidocaine. This patient was, however, excluded due to inadequate data entry regarding the inability to assess her pain intensity on their established scale [27].

Cardiotoxic effects have been described at higher levels (10–30 μg/mL) and include cardiac arrhythmias, heart block, myocardial depression, and cardiovascular collapse [33,34]. In other pediatric and adult studies, patients with seizure and cardiovascular disorders are generally excluded. The diagnosis of lidocaine toxicity, even in the case of severe adverse reactions to systemic lidocaine, is a clinical diagnosis as the levels can take time to result [35]. However, some authors have recommended monitoring lidocaine levels during continuous lidocaine infusions to avert these toxic effects.

Lipid emulsions have been used to lower lidocaine plasma levels in the event of toxicity [36]. The studies described here suggest that initiating a lidocaine infusion requires dose titrations and close monitoring for side effects [37,38]. Table 3 summarizes dosing considerations and pharmacokinetics for intravenous lidocaine that have been described in both adult and pediatric literature [22,23,24,25,26,32,35,36,37,38,39]. The pharmacokinetics of lidocaine can vary depending on several factors, including ethnicity, organ dysfunction, obesity, and concomitant opioid use [25]. The active metabolite of lidocaine, MEGX, is formed from the *N*-deethylation of lidocaine, which is mediated by cytochrome P-450 (CYP) enzymes CYP1A2 and CYP3A4 [40,41]. This indicates that lidocaine has the potential to interact with other drugs metabolized by the same enzymes.

## 3. Systemic Lidocaine Infusions in Patients with Cancer

Cancer-related pain can manifest in many ways and is frequently chronic. When opioids no longer have the desired effect or contribute to serious unwanted effects, other pain relief options to pursue include sedatives, ketamine infusions, and a regional anesthetic of a neural blockade, as described previously. Acknowledging that refractory pain is distressing for many reasons, including psychological well-being and quality of life, appropriate care may include the use of these other options for cancer-related pain. Apart from these adjuvants, bolus and continuous infusions of intravenous lidocaine have also been demonstrated to be efficacious in pediatric patients with cancer [32].

A retrospective study at St. Jude Children’s Research Hospital by Anghelescu et al. reviewed 29 hematology and oncology patients who received a total of 78 intravenous lidocaine infusions over a 10-year period [24]. Pain scores were reported as significantly lower each day during the first three days of administration. The infusions were generally well-tolerated, with only one patient developing orthostatic hypotension as described previously.

One case series described four pediatric patients with advanced-stage solid tumors with histories of multi-modal pain treatments who were given intravenous lidocaine [42]. Three of these patients experienced side effects associated with lidocaine, including changes in vision, hallucinations, and paresthesia, all of which resolved either spontaneously or with a decrease in the infusion rate. All four patients had a significant decrease in their pain scores four hours into the infusion, and even after the infusions were complete, the pain scores remained lower than the scores at the start of the infusion. Serum lidocaine levels did not correlate with infusion rates or toxicity. Another study evaluated the efficacy of systemic lidocaine infusions used in thirteen adult patients for neuropathic pain from a variety of underlying diagnoses. In that study, the dose-response curve was found to be very steep, showing that even slight increases in dose provided very large increases in pain relief [43]. The authors also noted that the serum concentration of lidocaine at the point where patients reached analgesia was variable. These studies suggest that lidocaine levels do not correlate with therapeutic efficacy or toxicity, and there is significant inter-patient variability in the dose required for analgesia, similar to opioid use in pediatric patients [44].

In pediatric oncology, new therapies can present new side effects—patients with high-risk neuroblastoma now have the benefit of being treated with anti-GD2 targeted therapies. However, due to the nature of the drug’s target—nervous tissue—adequate pain control can be difficult to achieve. A known side effect of this antibody is pain, typically in the extremities, back, and abdomen [45,46]. Commonly during these infusions, intravenous opioids are used prophylactically and for breakthrough pain relief, but some patients do not tolerate opioids at the doses needed to achieve analgesia, and some have refractory pain regardless. A case series evaluating a subset of five patients with neuroblastoma compared intravenous lidocaine to intravenous morphine during the anti-GD2 antibody infusion [45]. During the antibody treatment, if any of the patients experienced breakthrough pain, they were treated with an intravenous morphine bolus. While no difference in pain scores was found between the two groups, there was a decrease in morphine requirements and total morphine consumption, as well as a decrease in pain-related mobility impairment in the group receiving lidocaine. These results suggest that intravenous lidocaine can act as a potential adjunct in patients who experience significant opioid toxicity at analgesic doses.

A recent review was published evaluating articles with a combined seventy-three total patients who were treated with lidocaine parenterally for their complex, cancer-related pain [47]. Among the articles included in the review, there was no consensus on validated outcome measures that can be used to compare pain between treatment groups. However, each article did conclude that the patients who received parenteral lidocaine experienced pain relief to some degree. In the largest cohort study in the group, the authors concluded that no significant adverse events could be attributed to lidocaine [48]. The articles included in this review were reflective of studies or cases of both adult and pediatric patients with cancer-related pain.

For many patients with a cancer diagnosis, surgical procedures are a guaranteed part of their treatment plan and a common cause of pain. Pain management in the postoperative period is typically multi-modal and dependent on the type of surgery, patient characteristics, and patient goals. Systemic lidocaine has been used in a multi-modal approach in adults post-operatively with promising results [49,50,51]. There are few published studies evaluating the pediatric response to systemic lidocaine as postoperative analgesia. One retrospective analysis analyzed 50 patients recovering from a variety of surgeries [52]. Lidocaine was given as a continuous infusion to test safety and tolerability. Out of 50 patients, 11 reported side effects, but there was no conclusive, standardized way to document the grade or clinical significance of these effects.

Lastly, advanced-stage cancer can cause considerable, refractory pain. In one case report of a young girl with end-stage, metastatic, recurrent retinoblastoma, the providers found that despite intravenous opioid rotation, the patient had refractory bone pain [53]. She was receiving comfort measures, but her pain persisted to the point where she was no longer able to interact with her family. She was admitted, and after trials of many different analgesics with resulting unwanted sedation, she was given continuous intravenous lidocaine titrated to 35 μg/kg/min. Other pain medications were discontinued as she had improved pain control and mental status. Intravenous lidocaine provided appropriate pain relief for her and was up-titrated as needed with no limiting side effects.

## 4. Discussion

Refractory pain is a therapeutic dilemma frequently encountered in the pediatric cancer setting. The existing literature suggests that the use of intravenous lidocaine infusions in pediatric patients is safe and effective in treating cancer-related pain—particularly as an opioid-sparing adjuvant.

There were several limitations to the review of the existing literature on children. Most of the studies were either case reports or retrospective studies, and only one prospective, randomized, double-blinded study was performed among pediatric patients. Another challenge is the lack of a standardized and unified way to measure pain. One review of the literature recognized that none of the articles analyzed used a validated measure of pain [47]. While each of the mentioned articles acknowledges that lidocaine did effectively reduce pain scores in whichever way they were measured, it is important moving forward to have an established and validated measure of pain, especially in pediatric patients who may not be able to verbalize their discomfort. Validated pain measurements for pediatrics do exist [54], but when different scales are used, it is difficult to compare pain scores between institutions.

Randomized, controlled clinical trials would also help establish an official recommended dose in pediatric patients to assess the analgesic effect and the safety of lidocaine. The primary outcome measure of paramount importance is a validated measure of pain. Secondary outcomes important to capture include adverse events and the change in the quality of life after the use of lidocaine infusions. Multicenter studies could increase the power of statistical analysis to effectively measure and analyze pain reduction and adverse effects.

## 5. Conclusions

Refractory pain is a therapeutic dilemma frequently encountered in the pediatric cancer setting. The careful selection of an analgesic regimen should be based on individual patient factors, including the level of pain, response to therapy, and physiologic profile. The existing literature suggests that the use of intravenous lidocaine infusions in pediatric patients is safe and effective in treating cancer-related pain—particularly as an opioid-sparing adjuvant. Future prospective studies and randomized controlled trials comparing current practices with lidocaine infusions as adjuvant analgesic therapy in the pediatric oncology population would further elucidate the role of systemic lidocaine in this setting.

## Figures and Tables

**Table 1 children-09-01934-t001:** Indications for systemic lidocaine infusion.

Surgical	Medical
Alternative to regional anesthesia	Opioid dependence or tolerance
Epidural—contraindicated, inadequate, or failed	Opioid-induced hyperalgesia
Laparoscopic surgery	Accrual or at increased risk of opioid side-effects
Spine surgery and limb amputations	Acute neuropathic pain
Surgery at a site of chronic pain	Acute nociceptive pain
Previous experience with poorly controlled pain	Chronic neuropathic pain
Enhanced recovery protocols	Chronic nociceptive pain
Prevention or treatment of ileus	Mixed/multiple
Trauma	Cancer-related pain, status migraine
Multiple significant injuries, burns, degloving, crush injury, rib, clavicle, or sternal fractures	

Adapted with permission from ref. [25]. 2022 Elsevier.

**Table 2 children-09-01934-t002:** Contraindications for systemic lidocaine infusion.

Sensitivity or allergy to lidocaine
Significant heart disease, severe cardiac failure, or heart block of any degree
History of or active dysrhythmia
Concurrent treatment with Class I antiarrhythmics or amiodarone use < 3 months
Severe hepatic impairment
Severe renal impairment
History of or active seizure disorder
History of malignant hyperthermia
Concurrent use of regional anesthesia or use within 4–8 h
Acute porphyria

Adapted with permission from ref. [25]. 2022 Elsevier.

**Table 3 children-09-01934-t003:** Intravenous lidocaine pharmacokinetics.

Dosing for analgesia	1–2 mg/kg bolus over 30 min, followed by a continuous infusion of 0.5–2 mg/kg/h
Onset of action	45–90 s
Duration	10–20 min
Half-life	1.5–8 h ^1^ (parent drug)
Active metabolites	MEGX ^2^: may cause toxicity in heart failure patients (2 h half-life)
	GX ^3^: may accumulate in patients with renal failure (10 h half-life)
Therapeutic plasma concentrations	1.5–5 μg/mL at steady state
Adverse reactions	Dizziness, tinnitus, QRS prolongation, sinus slowing, hypotension, dysrhythmias, seizures, methemoglobinemia, malignant hyperthermia, anaphylactoid reactions

Adapted with permission from ref. [22]. 2022 WILEY ^1^ Half-life may be prolonged in patients with heart failure, liver dysfunction, or renal dysfunction. ^2^ MEGX = Monoethylglycinexylidide. ^3^ GX = Glycinexylidide.

## Data Availability

Not applicable.

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
