# Peer review of "Systemic Lidocaine Infusions for Pediatric Patients with Cancer-Related Pain"

_children, 2022, doi:10.3390/children9121934_

Round 1
Reviewer 1 Report
The manuscript titled “Systemic Lidocaine Infusions for Pediatric Patients with Cancer-Related Pain” is a review that discusses the use of systemic lidocaine infusions for analgesia in pediatric oncology patients with cancer-related pain.
The authors could clarify the following point.
The author uses indistinct units, for example, hr or hours, unify in the manuscript.
Line 109 changes K by Kg.
The sentence in lines 170-172 is difficult to understand. Could you improve the redaction?
The authors mention the pharmacokinetic of lidocaine in table 3. Is so important that they mention in what type of population were do realize the pharmacokinetic studies, for example, Caucasian, mestizo, and African-American people. Also, is important to indicate which enzyme of CYP450 is responsible for the metabolism of lidocaine. Also is interesting that the author indicates if lidocaine had shown drug interactions with other drugs.
The sentence inline 230-237 is complicated to read. Could you improve the redaction, please?
Actually, there are some tests validated that let evaluated the pain in pediatric patients. I am struck by the fact that the authors indicate that tests are needed that allow the evaluation of pain in pediatric patients, mainly when it is difficult for them to communicate. If I remember correctly, there are several tests that can assess pain in unconscious patients.
Author Response
Dear Reviewer,
Thank you for your thoughtful comments and recommendations. We have incorporated the following changes based on your feedback:
- We have unified the words for units throughout.
- We corrected the typographic error of “K” to Kg in line 109.
- Improvements were made to the sentence structure to provide clarity in lines 170-172 and 230-237.
- A clarification was made to note that there are validated pain measurement scores in both pediatric and unconscious patients. However, when studies use different pain measurement scores it becomes difficult to compare the efficacy of the treatment without a unified way of measuring.
- In discussion of the pharmacokinetics of lidocaine, we included the CYP450 enzymes responsible for lidocaine metabolism and the potential for drug-drug interactions. We acknowledged that the pharmacokinetics of lidocaine could vary dependent on factors including ethnicity of the patients, obesity, and other medication use.
We appreciate your review.
Reviewer 2 Report
The undertaken research is part of the discussions on reducing pain.
The presented material is a valuable overview, and the announcement
of further research deserves recognition.
Author Response
Dear Reviewer,
Thank you for your positive feedback. In the interest of due diligence and to be thorough, we have the made the following changes to improve the paper and make it more robust:
- We corrected a typographical error of citation “24,26” to “24-26” on line 105.
- We have clarified the potential for cardiotoxic effects of lidocaine and commented on the current literature regarding recommendations for obtaining lidocaine levels in the setting of toxicity.
We appreciate your review.